# Longitudinal Analysis of Antiphospholipid Antibody Dynamics after Infection with SARS-CoV-2 or Vaccination with BNT162b2

**DOI:** 10.3390/ijms24010211

**Published:** 2022-12-22

**Authors:** Manca Ogrič, Polona Žigon, Snezna Sodin-Semrl, Mirjana Zlatković-Švenda, Marija Zdravković, Milica Ovuka, Tinka Švec, Katja Lakota, Peter Radšel, Žiga Rotar, Saša Čučnik

**Affiliations:** 1Department of Rheumatology, University Medical Centre Ljubljana, 1000 Ljubljana, Slovenia; 2Faculty of Mathematics, Natural Sciences and Information Technologies, University of Primorska, 6000 Koper, Slovenia; 3Institute of Rheumatology, Faculty of Medicine, University of Belgrade, 11000 Belgrade, Serbia; 4Faculty of Medicine Foča, University of East Sarajevo, 73300 Foča, Bosnia and Herzegovina; 5University Hospital Medical Center Bezanijska kosa, 11080 Belgrade, Serbia; 6Faculty of Medicine, University of Belgrade, 11000 Belgrade, Serbia; 7General Hospital Pančevo, 26000 Pančevo, Serbia; 8Department of Intensive Internal Medicine, University Medical Centre Ljubljana, 1000 Ljubljana, Slovenia; 9Faculty of Medicine, University of Ljubljana, 1000 Ljubljana, Slovenia; 10Faculty of Pharmacy, University of Ljubljana, 1000 Ljubljana, Slovenia

**Keywords:** COVID-19, SARS-CoV-2, BNT162b2 vaccine, autoantibodies, antiphospholipid antibodies, healthcare professionals, APS

## Abstract

Antiphospholipid antibodies (aPL) comprise a group of autoantibodies that reflect prothrombotic risk in antiphospholipid syndrome (APS) but may also be present in a small proportion of healthy individuals. They are often transiently elevated in infections, including SARS-CoV-2, and may also be associated with vaccine-induced autoimmunity. Therefore, we aimed to investigate the dynamics of aPL in COVID-19 patients and in individuals (healthcare professionals—HCPs) after receiving BNT162b2 vaccine and to compare aPL levels and positivity with those found in APS patients. We measured solid-phase identifiable aPL, including anticardiolipin (aCL), anti-β2 glycoprotein I (anti-β2GPI), and anti-prothrombin/phosphatidylserine (aPS/PT) antibodies in 58 HCPs before and after vaccination (at 3 weeks, 3, 6, and 9 months after the second dose, and 3 weeks after the third booster dose), in 45 COVID-19 patients hospitalized in the ICU, in 89 COVID-19 patients hospitalized in the non-ICU (at admission, at hospital discharge, and at follow-up), and in 52 patients with APS. The most frequently induced aPL in COVID-19 patients (hospitalized in non-ICU) were aCL (50.6% of patients had positive levels at at least one time point), followed by anti-β2GPI (21.3% of patients had positive levels at at least one time point). In 9/89 COVID-19 patients, positive aPL levels persisted for three months. One HCP developed aCL IgG after vaccination but the persistence could not be confirmed, and two HCPs developed persistent anti-β2GPI IgG after vaccination with no increase during a 1-year follow-up period. Solid-phase aPL were detected in 84.6% of APS patients, in 49.4% of COVID-19 patients hospitalized in the non-ICU, in 33.3% of COVID-19 patients hospitalized in the ICU, and in only 17.2% of vaccinated HCPs. aPL levels and multiple positivity were significantly lower in both infected groups and in vaccinated individuals compared with APS patients. In conclusion, BNT162b2 mRNA vaccine may have induced aPL in a few individuals, whereas SARS-CoV-2 infection itself results in a higher percentage of aPL induction, but the levels, persistence, and multiple positivity of aPL do not follow the pattern observed in APS.

## 1. Introduction

In 2019, a novel betacoronavirus, severe acute respiratory syndrome coronavirus 2 (SARS-CoV-2), caused a pandemic of coronavirus disease 2019 (COVID-19). COVID-19 manifests in very different forms: from the asymptomatic form to flu-like disease to the severe form of the disease with pneumonia with acute respiratory distress syndrome and cytokine storm [1,2]. On 11 December 2020, the FDA announced the first recommendation for a vaccine against COVID-19, the mRNA vaccine BNT162b2 (Pfizer-BioNTech) [3], which was later approved for the mitigation and prevention of severe COVID-19.

Several scientific groups have reported the occurrence of antiphospholipid antibodies (aPL) in patients with COVID-19 and proposed the possibility of antiphospholipid syndrome (APS) triggered by the SARS-CoV-2 virus [4,5,6,7]. APS is a systemic autoimmune thromboinflammatory disorder characterized clinically by a predisposition to arterial, venous, or microvascular thrombotic events, as well as pregnancy complications and serologically by the persistent presence of aPL. Laboratory criteria for definite APS [8] include three subgroups of aPL: lupus anticoagulants (LA), anticardiolipin (aCL), and anti-β2 glycoprotein I (anti-β2GPI) of the immunoglobulin (Ig) G and M classes. Several studies also confirmed the association of non-criteria antibodies with APS, such as prothrombin-phosphatidylserine (aPS/PT) and also aCL and anti-β2GPI of the IgA class, but their inclusion in the APS classification criteria remains controversial [9,10,11]. To avoid false-positive tests due to infections, positive aPL tests should be repeated at an interval of at least 12 weeks as it is known that aPL can transiently occur during various infections, including skin infections (18%), human immunodeficiency virus infection (17%), pneumonia (14%), hepatitis C virus (13%) and urinary tract infections (10%) [12].

One of the first detailed analyses of 23 studies investigating aPL levels in COVID-19 comprising a total of 250 patients reported that LA, aCL, and anti-β2GPI were present in 64%, 9%, and 13% of cases, respectively [13], with IgM antibodies as the most common isotype. In contrast, a study comparing moderate and severe forms of COVID-19 disease found that aCL IgG levels were highly and independently associated with disease severity [14]. Importantly, none of these studies reported retesting aPL on a second occasion; thus, it is not clear whether the aPL presence in COVID-19 patients was transient or persistent. The only study that repeated aPL testing after one month included 31 patients with COVID-19 and found elevated aPL levels in 23 (74%) patients, of whom 21 (67%) had LA, 7 had aCL and aPS/PT, and 3 had anti-β2GPI [15]. Their main finding was that 9 out of 10 retested LA positive patients were negative at the second testing. This observation supports the frequent single and transient LA positivity during the acute phase of COVID-19 infection. A large meta-analysis, comprising 21 studies and 1159 patients, published in April 2021, showed that nearly half of patients with COVID-19 were positive for at least one of the aPL. The most frequently reported aPL was LA. aPL were significantly more frequently reported in critically ill patients, and aPL were not significantly associated with disease outcomes such as venous thrombosis, invasive ventilation and mortality [16].

Another literature review, published a few months later, included 34 studies with a total of 3288 COVID-19 patients and found that 547/3288 (16.6%) cases were aPL positive (including LA) [17]. In another review that excluded single-case studies, the incidence of reported aPL positive cases (including LA) was found to be 33%, with an interquartile range (IQR) of 11 to 52% [18]. Most of the higher incidence group was due to the presence of LA, sometimes reported in >80% of cases tested. The reported incidence of solid-phase identified aPL (i.e., aCL, anti-β2GPI, aPS/PT) was generally lower. In addition, most identified aPL were of fairly low titer, and multiple aPL positivity was rarely reported. Thus, double and triple positivity were found in only a few individuals. The authors emphasized that repeat tests for persistence of aPL were rarely performed or described, and when they were reported, the authors suggested the identified aPL were transient in nature. Such transient aPL do not indicate an autoimmune disease in the classic sense of APS.

Theoretically, a scenario similar to SARS-CoV-2 infection could play out after vaccination. Vaccine-associated autoimmunity is a well-known phenomenon related to cross-reactivity between certain pathogenic elements present in the vaccine and specific human proteins [19,20,21]. For newly developed mRNA vaccines against COVID-19, in addition to molecular mimicry, binding of mRNA to pattern recognition receptors has been described as another possible mechanism, leading to activation of various proinflammatory cascades known to underlie various immune-mediated diseases [22]. Initial trials confirmed the safety of the mRNA vaccine [23], but due to the short development timelines, active investigation of potential adverse effects, including autoimmune reactions, is of utmost importance.

To date, several cross-sectional studies have examined autoantibody profiles in sera from COVID-19 patients and a few in sera from vaccinated individuals, but studies comparing these groups with the APS group are lacking. Our main aim was to longitudinally investigate the dynamics of aPL. We investigated the induction and persistence of aPL in COVID-19 patients and in individuals (healthcare professionals—HCPs) after receiving BNT162b2 vaccine and compared the levels, percentage of positive samples, and multiple positivity among four groups: HCPs, COVID-19 patients hospitalized in the ICU, COVID-19 patients hospitalized in the non-ICU, and APS patients.

## 2. Results

The main demographic characteristics of the four participant groups (group I—HCPs, group II—COVID-19 patients hospitalized in the ICU, group III—COVID-19 hospitalized in the non-ICU, and group IV—APS patients) are shown in Table 1. Both COVID-19 patient groups are significantly older compared with the HCP and APS groups.

### 2.1. The Induction of aPL during Infection with SARS-CoV-2

The induction of aPL positivity was investigated in group III (COVID-19 patients hospitalized in the non-ICU) at three time points (admission, hospital discharge, and 3-month follow-up after hospital discharge). The percentages of positive samples for all aPL are presented in Table 2, the longitudinal trends are shown in Table 3, and the longitudinal analysis of levels is shown in Figure 1. 

Longitudinal analysis revealed that the percentage of positive aCL IgG and IgM samples and their levels increased during hospitalization (*p* < 0.0001 for both) and decreased at follow-up (*p* = 0.01 and *p* < 0.0001, respectively), although levels were still higher at follow-up than at admission (both *p* = 0.003). aCL IgA levels and the percentage of positive samples decreased at follow-up (*p* < 0.0001) (Table 2, Figure 1). A total of 45/89 (50.6%) of COVID-19 patients had at least one positive aCL (G, M, and/or A) during the observation period. In 8/45 (17.8%), the positive levels persisted at the last visit. The predominant trend in aCL dynamics observed in 25/45 (55.6%) patients was an increase in levels during hospitalization and a decrease at follow-up (Table 3).

Statistically, anti-β2GPI levels and the percentage of positive samples did not change between the time of admission, hospital discharge, and follow-up. Among the patients, 19/89 (21.3%) had at least one positive anti-β2GPI (G, M, and/or A) (Table 2 and Table 3 and Figure 1). The predominant trend in anti-β2GPI dynamics observed in 8/19 (42.1%) patients was a decrease in positive levels during hospitalization (Table 3). During our observation period, anti-β2GPI were induced in 3/19 (15.8%) patients, of whom positive levels persisted in 2/19 (10.5%) (Table 3).

The aPS/PT levels of IgG and IgA remained the same during hospitalization but decreased at follow-up (*p* = 0.03 and *p* < 0.0001, respectively), whereas IgM levels did not change. During the observation period, 7/89 (7.9%) patients had at least one positive aPS/PT (IgG, IgM, and/or IgA), but aPS/PT were induced in only one patient during hospitalization but positive levels persisted in none (Table 3).

Comparing different subtypes of aPL that were increased at least at one time point during the observation period, we found that aCL were present in the highest percentage (50.6%), followed by anti-β2GPI (21.3%) and aPS/PT (7.9%). Importantly, we found nine COVID-19 patients with persistent aPL positivity. Notably, seven had positive aCL IgG, and one patient was found with positive anti-β2GPI IgG, and one patient with double aCL and anti-β2GPI positivity.

### 2.2. The Induction of aPL in HCPs after Vaccination

The induction of aPL positivity was investigated in 58 HCPs before vaccination with BNT162b2 and at different time points after the second dose (3 weeks (n = 58), 3 months (n = 55), 6 months (n = 50) and 9 months (n = 45) after vaccination) and after the third-booster dose (n = 33). All the results are presented in Table 2 and in Figure 2.

Before vaccination, one HCP (1.7%) had a positive aCL IgM level and none had aCL IgG or IgA. After vaccination, none of the HCPs developed aCL IgM or IgA antibodies, but one HCP (1.7%) had positive aCL IgG levels 6 months after vaccination; however, the positivity could not be confirmed 12 weeks later because of missing samples (Table 2 and Table 4, Figure 2).

Positive levels of anti-β2GPI IgG antibodies were detected in 5/58 (8.6%) HCP serum samples before vaccination. The levels did not increase after vaccination. After vaccination, positive anti-β2GPI IgG titers were newly detected in the sera of two HCPs, and their elevated levels persisted without significant increase during the 1-year follow-up period (Table 2 and Table 4, Figure 2).

Before vaccination, one (1.7%) HCP had positive levels of anti-β2GPI IgM and one (1.7%) HCP had positive levels of anti-β2GPI IgA. After vaccination, none of the previously negative HCPs developed positive levels (Table 2, Figure 2).

We did not detect positive levels of aPS/PT (IgG, IgM, or IgA) before or after vaccination in any of the HCP (Table 2, Figure 2).

Overall, in our group of 58 HCPs, 7/58 (12.1%) of HCPs were positive for at least one of the aPL tested in our study before vaccination; in particular, 6/7 were positive for a single aPL before vaccination, and 1/7 was double aPL positive (Table 2). After vaccination, an additional three HCPs were identified with single aPL positivity, two of whom remained positive on subsequent testing (Table 4).

In total, 3/51 (5.9%) previously negative HCPs developed aPL after vaccination, with persistent positivity detected in the sera of two HCPs (3.9%). None of the HCPs who had positive levels of aPL were infected with SARS-CoV-2 before vaccination.

### 2.3. The Prevalence of aPL in Vaccinated HCPs, COVID-19 and APS Patients

The number and percentage of positive aPL are shown in Table 2 for four groups of patients: HCPs (group I), COVID-19 patients hospitalized in the ICU (group II), COVID-19 patients hospitalized in the non-ICU (group III, time point of hospital discharge), and APS patients (group IV).

We found that the percentage of positive aPL was highest in the APS group, followed by the COVID-19 groups, and was lowest in the HCP group (Table 2). This trend was observed for aCL IgG, aCL IgM, anti-β2GPI IgA, and aPS/PT IgA. Similarly, for anti-β2GPI IgG, anti-β2GPI IgM, aPS/PT IgG, and aPS/PT IgM, the percentage of positive samples was highest in APS patients but comparable in HCPs and COVID-19 patients.

Importantly, multiple aPL positivity was observed much more frequently in APS patients (32.7% had triple-positive aPL) compared with COVID-19 patients (only 2.2% triple-positive aPL samples in the COVID-19 group II and 1.1% triple-positive aPL samples in the COVID-19 group III) and HCPs (no HCPs with triple-positive aPL). Similar results were found for double positivity, whereas single positivity in COVID-19 group III at hospital discharge was similar to that in the APS group (Table 2). At least one positive aPL was found in 17.2% of the HCPs (group I), 33.3% of the COVID-19 group II patients, 49.4% of the COVID-19 group III patients, and 84.6% of the APS group.

### 2.4. The aPL Levels in Vaccinated HCPs, COVID-19 and APS Patients

Using the multiple comparison test, we found statistically significant differences in the levels of all tested aPL between APS patients and HCPs (Figure 3). Differences were also found between APS and the COVID-19 group II for all aPL (except anti-β2GPI IgA and aPS/PT IgA) and between APS and the COVID-19 group III (time of hospital discharge) for all except aCL IgG and IgM. Differences between two COVID-19 groups were found for aCL IgM, anti-β2GPI IgA, aPS/PT IgG, and aPS/PT IgA. Differences between HCPs and the COVID-19 group II patients were found for aCL IgA, anti-β2GPI IgA, aPS/PT IgG, and aPS/PT IgA, and between HCPs and COVID-19 group III patients for aCL IgG, IgM, and IgA.

To illustrate the data graphically, Figure 4 presents the results in the form of a heatmap. Note that patients with APS have a significantly higher frequency of aPL positivity as well as significantly higher levels of aPL compared with COVID-19 patients and healthy vaccinated HCPs.

### 2.5. Clinical Manifestations Related to APS in Vaccinated HCPs and COVID-19 Patients

None of the HCPs who had positive levels of aPL experienced any thrombotic adverse events during the 1-year observation period.

Two COVID-19 patients (group II) hospitalized in the ICU had pulmonary embolism, ten had venous thrombosis, and four had both pulmonary embolism and deep venous thrombosis. Six of them had positive levels of aPL on one occasion. One patient, triple positive for aCL, anti-β2GPI and aPS/PT, died.

Four COVID-19 patients hospitalized in the non-ICU (group III) had a history of arterial thrombosis, three had a history of venous thrombosis, and two had a history of obstetric complications relevant to the APS classification criteria. None of these nine patients were classified as an APS patient, and none of them had positive aPL values twice during our observation. Moreover, none of them experienced a new thrombotic event during our observation. One patient experienced arterial thrombosis and microthrombosis during hospitalization in the non-ICU, was triple aPL positive at admission and remained double positive at hospital discharge and 3-month follow-up. This patient thus met both clinical and laboratory criteria for definite APS.

## 3. Discussion

In the years following the outbreak of the COVID-19 pandemic, there have been many studies examining the induction of aPL in COVID-19 disease, whereas studies investigating the persistence of aPL and the induction of aPL after vaccination against COVID-19 are scarce. Therefore, we performed a longitudinal investigation of aPL dynamics in COVID-19 patients with a 3-month follow-up and in vaccinated HCPs with a 1-year follow-up.

Thromboembolic events in COVID-19 patients are observed, which has prompted researchers to investigate aPL in COVID-19 patients [24,25,26]. In some reports, more than 50% of hospitalized patients with COVID-19 had autoantibodies of any type [27]. Zhang et al. [28] were the first to report the presence of aPL in patients with COVID-19, and IgA was the most common isotype, which is also observed in our results (severely ill COVID-19 patients hospitalized in the ICU– group II), where patients had significantly elevated levels of anti-β2GPI IgA and aPS/PT IgA compared with HCPs. In a recent review article [29], the authors noted that the positivity of any aPL in COVID-19 patients ranged from 5 to 71%, showing wide variability in results due to different study cohorts, sample timing, and methods used. We found that in group II comprising COVID-19 patients from the ICU, 33.3% of patients had at least one positive aPL, whereas in the group III comprising COVID-19 patients from the non-ICU, 49.4% of patients had at least one positive aPL. Surprisingly, the group II comprising severely ill COVID-19 patients had a lower percentage of positive samples than the COVID-19 patients in group III, with two exceptions: anti-β2GPI IgA and aPS/PT IgA. This observation could be partially explained by the difference in thrombosis rate between the groups, as 16/45 (35.6%) of COVID-19 patients hospitalized in the ICU had thrombosis compared with 1/89 (1.1%) of COVID-19 patients hospitalized in the non-ICU. The loss of positivity of aPL has been observed in clinical practice immediately after thrombosis in APS patients, and it is suggested that if it occurs at the exact time of thrombosis, it may be due to loss by deposition in the thrombosis [30].

It is important to emphasize that many published studies have not examined a very important aspect of aPL, namely the significance of their persistent positivity. It is well documented that aPL may occur transiently during various infections and fluctuate over time; therefore, only persistent positivity is associated with an increased risk of thromboembolic events significant for APS. In most studies examining aPL in COVID-19 patients, levels were measured only once, so their conclusions are questionable. However, Espinosa et al. investigated aPL twice at 12 weeks apart and reported that aPL positivity at low titers persisted in only half of the COVID-19 patients studied [31]. Recently another group of investigators examined the persistence of aPL positivity after COVID-19. Of the 45 aPL positive patients who were followed up at 12 weeks, 13 patients (28.9%) had at least one persistent aPL with a single positivity in 69.2%, double positivity in 15.4%, and triple positivity in 15.4% [32]. In our study, we measured aPL at three different time points: at admission, hospital discharge (median 10 days (IQR 8–13 days)) and at 3-month follow-up after hospital discharge. We found some important results. First, the most common trend observed for aCL was an increase in levels at hospital discharge and a decrease in levels at follow-up (55.6% of samples positive at at least one time point had this trend). Second, the levels of anti-β2GPI were highest at admission and decreased at hospital discharge (42.1% of samples positive at at least one time point had this trend). Third, at 3-month follow-up, the levels were still positive in 8/45 (17.8%) of positive aCL patients and in 2/19 (10.5%) of positive anti-β2GPI patients.

Several studies have shown that the risk of thrombosis increases with each additional aPL detected [33,34,35]. Based on these findings, there are some suggestions to consider only patients with triple positivity as definite APS [33]. As noted in a recent review, multiple positivity for aPL has rarely been reported in studies investigating aPL in COVID-19 [18]. Thus, double or triple positivity has been found in only a few isolated individuals. The results of our study support these findings, as multiple aPL positivity was observed much more frequently in APS patients (32.7% had triple-positive aPL) than in COVID-19 patients (there were only 2.2% triple-positive aPL samples in the COVID-19 group II and 1.1% triple-positive aPL samples in the COVID-19 group III) and in HCPs (no individuals with triple-positive aPL). Similar results were found for double positivity. The results of our study clearly show that patients with APS have significantly higher autoantibody levels and a higher percentage of positivity compared with COVID-19 patients. Although APS usually affects young people because the first vascular event usually occurs in young adults and rarely in people older than 60 years, the frequency of aPL positivity in the general population is known to increase with age [36]. Therefore, the different age distribution of our patient groups (COVID-19 patients were older, whereas APS patients were younger) not only reflects the etiologies of these conditions and a possible bias in our study, but also confirms that the frequency of aPL positivity is much higher in APS patients compared with COVID-19 patients, although the APS group was the youngest one in this study.

In addition, we investigated the induction of aPL after vaccination with BNT162b2. In our study, there was no clear trend of induction or increase of aPL. We found that one HCP developed aCL IgG after vaccination, although persistence was unknown, and two HCPs developed persistent anti-β2GPI IgG positivity. The levels persisted equally positive at the 1-year follow-up, but neither experienced any adverse clinical events.

Our results support, to some extent, the findings of other studies in this field. Several authors have studied the induction of autoantibodies after BNT162b2 vaccination, and in most cases no significant induction of autoantibodies was observed. Borghi et al. published that vaccination with BNT162b2 and also ChAdOx1 did not induce early autoantibody production [37], Noureldine et al. found no clear pattern of an increase in aPL titers, with the exception of one female participant who had a significant increase in aPL IgM levels after each dose [38], and Thurm et al. concluded that vaccines do not significantly promote the occurrence of autoantibodies, which are commonly associated with various systemic autoimmune diseases [39]. Importantly, the few isolated cases in which autoantibodies developed after vaccination are not sufficient to establish a link between vaccination with BNT162b2 mRNA vaccine and autoimmune markers. The frequency of occurrence of antibodies is very low, so it is unlikely to exceed the frequency of occurrence of antibodies in the general population. Nevertheless, the occurrence of aPL in these individuals may be overestimated because of the high frequency of measurement in our study and the long follow-up period as a higher proportion of aPL may be associated with other causes such as underlying infection, stress, or injury. On the other hand, it is also possible that this study underestimates the true incidence because of the small sample size.

However, some previous studies investigating other vaccines documented an increase in autoantibodies and the occurrence of autoimmune diseases following vaccination with human papillomavirus [40,41,42], influenza [43], and hepatitis B (HBV) [44]. These results should be interpreted with caution as a meta-analysis published later found no association between HPV vaccination and autoimmunity [45]. There are a few recent studies, mostly case-control studies, reporting induction of various autoimmune diseases in individuals after vaccination with BNT162b2, including rheumatoid arthritis [46], ANCA-associated vasculitis [47,48], and microscopic polyangiitis [49].

The main strengths of this study are the long duration and high follow-up rate in vaccinated HCPs and the longitudinal data of aPL trends in COVID-19 patients with follow-up time point after 12 weeks. This is one of the rare studies of its kind in which samples of HCPs were studied before and after vaccination for up to one year to compare the profile and levels of aPL with those of COVID-19 and APS patients. Although this study included a rather small group of participants, we make important and similar conclusions about the association between SARS-CoV-2, COVID-19 vaccines and autoimmunity as previously reported.

Based on our findings, we can conclude that aPL induction after BNT162b2 vaccination occurs in a small number of individuals but not to the extent of infection and without clinical consequence, whereas SARS-CoV-2 infection induces aPL in a higher percentage of patients, with a small percentage of them having persistent aPL, but not to the extent seen in APS patients.

## 4. Materials and Methods

### 4.1. Study Design and Participants

This observational study was conducted at the Department of Rheumatology, University Medical Centre Ljubljana (UMCL), Ljubljana, Slovenia. Serum samples comprising four participant groups were collected in parallel at the UMCL and at the General Hospital (GH) Pančevo.

Group I comprised samples from HCPs (employees of the Division of Internal Medicine, UMCL), without self-reported rheumatic autoimmune disease. Samples were collected during the pandemic, before vaccination with BNT162b2, and at different time points after the second dose (3 weeks (n = 58), 3 months (n = 55), 6 months (n = 50), and 9 months (n = 45) after vaccination) and 3 weeks after the third booster dose (n = 33). Among the HCPs, 17/58 (29.3%) were infected before vaccination according to self-reports, and antibodies to SARS-CoV-2 (anti-spike and/or anti-nucleocapsid antibodies) were detected [50].

Group II comprised 45 COVID-19 patients with a severe form of the disease who were treated in the ICU at UMCL in Slovenia from May to December 2020 before the start of vaccination. Their serum samples were collected on days 8–12 of their hospitalization. Clinical manifestations significant for APS were monitored.

Group III comprised 89 COVID-19 patients hospitalized in the non-ICU at GH Pančevo from January to May 2021. Their serum samples were collected at admission, at hospital discharge (days 5–29) and at the 3-month follow-up after hospital discharge. Clinical data comprising history of clinical manifestations significant for APS were obtained and also monitored during the observation period.

Group IV comprised 52 patients with APS who were recruited from the Department of Rheumatology, UMCL between March and June 2019 before the pandemic. All the patients met the updated international classification criteria for APS.

Informed consent was obtained from all the subjects involved in the study. The study was conducted according to the guidelines of the Declaration of Helsinki and was approved by the Ethics Committee of the Republic of Slovenia (#0120-7/2019/5, #0120-422/2020/6 and #0120-113/2021/4) and by the Ethics Committee of the GH Pančevo (#01-1492/21).

### 4.2. Laboratory Tests

All samples were tested for the presence of aPL, including aCL, anti-β2GPI, and aPS/PT, of IgG, IgM, and IgA isotypes.

aPL were determined using in-house ELISAs according to previously described protocols, specifically aCL ELISA [51], anti-β2GPI ELISA, and aPS/PT ELISA [52]. Values above the 99th percentile of the healthy control population were considered positive, specifically for aCL ≥ 11AU, for anti-β2GPI ≥ 2 AU and for aPS/PT ≥ 5 AU.

### 4.3. Statistical Analyses

Statistical analyses were performed using GraphPad Prism 8. The normality of the continuous data distribution was tested using the Shapiro–Wilk test. Summary statistics are presented as medians and 25th–75th percentiles because of the non-normal distribution. For dependent groups, the Friedman test was used; for more than two independent groups, the Kruskal–Wallis test followed by the Dunn’s multiple comparison test was used; for categorical data, the chi-square test was used. A *p* value of less than 0.05 was considered statistically significant in all statistical analyses. A heat map (created in GraphPad Prism 8) of all test results was created for graphical representation of all the data.

## Figures and Tables

**Figure 1 ijms-24-00211-f001:**
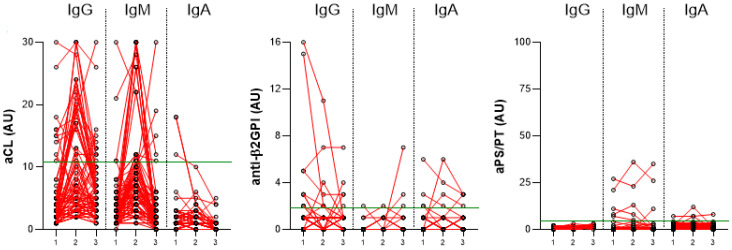
Longitudinal trends of aCL, anti-β2GPI, and aPS/PT, IgG, IgM, and IgA during infection with SARS-CoV-2 in COVID-19 patients hospitalized in the non-ICU (group III) at three time points (1—at admission, 2—at hospital discharge, 3—at 3-month follow-up after hospital discharge). The red lines connect the aPL levels of the individual COVID-19 patients at three time points. Values above the green line (99th percentile of healthy blood donors) are considered positive.

**Figure 2 ijms-24-00211-f002:**
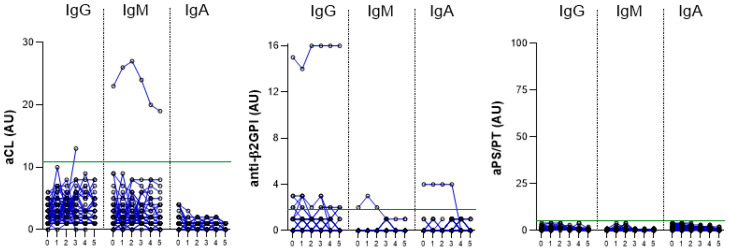
Longitudinal trends of aCL, anti-β2GPI, and aPS/PT, IgG, IgM, and IgA before (0) and after vaccination (time points: 1—3 weeks after vaccination, 2—3 months after vaccination, 3—6 months after vaccination, 4—9 months after vaccination, and 5—3 weeks after the third booster dose). The blue lines connect the aPL levels of the individual HCPs. Values above the green line (99th percentile of healthy blood donors) are considered positive.

**Figure 3 ijms-24-00211-f003:**
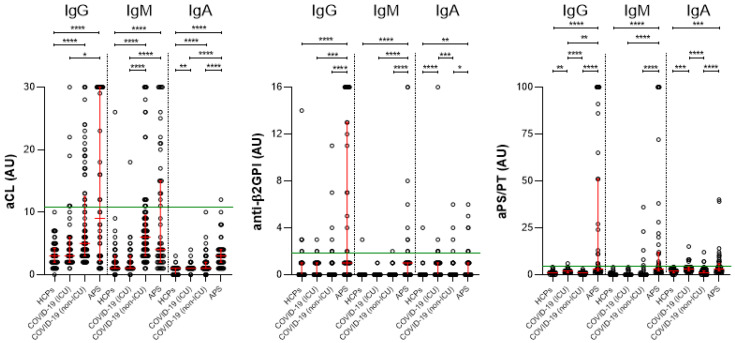
Levels of aCL, anti-β2GPI, and aPS/PT, IgG, IgM, and IgA in HCPs after vaccination (group I), COVID-19 patients hospitalized in the ICU unit (group II), COVID-19 patients hospitalized in non-ICU (group III) (at hospital discharge), and APS patients (group IV). Legend: Kruskal–Wallis test with Dunn’s multiple comparison test were used. *p*-value of <0.05 was considered as statistically significant. * *p* < 0.05; ** *p* < 0.01; *** *p* < 0.001; **** *p* < 0.0001. The red lines represent the median levels and IQR. Values above the green line (99th percentile of healthy blood donors) are considered positive.

**Figure 4 ijms-24-00211-f004:**
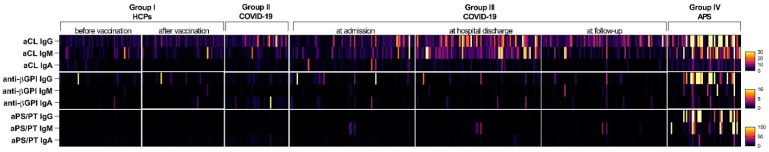
Heatmap showing all results of all participants, separated by tested aPL (vertical) and by groups (horizontal). The legend (colors) represents the titers of each aPL (black represents low levels; yellow represents high levels of each antibody). The results of the same patients are shown in the vertical line. Both HCP groups and all three time points of the COVID-19 group III have the same order of included participants. Therefore, they can be compared directly.

**Table 1 ijms-24-00211-t001:** Number of participants, sex and age distributions in the groups.

	Group I HCPs	Group II COVID-19 ICU	Group III COVID-19 Non-ICU	Group IV APS	*p*
No. of Participants	58	45	89	52	
Age (years)					
Median (IQR), (min–max)	46 (35–55) (24–64)	70 (58–78) (42–85)	62 (52–67) (27–78)	39 (34–56) (23–80)	*p* < 0.0001 * HCPs-COVID-19 group II: *p* < 0.0001, HCPs-COVID-19 group III: *p* < 0.0001, HCPs-APS: ns, COVID-19 group II-APS: *p* < 0.0001. COVID-19 group III-APS: *p* < 0.0001. COVID-19 group II–COVID-19 group III: *p* = 0.02.
Sex (n)					
Female	45	8	31	36	*p* < 0.0001 ** HCPs-COVID-19 group II: *p* < 0.0001, HCPs-COVID-19 group III: *p* < 0.0001, HCPs-APS: ns, COVID-19 group II-APS: *p* < 0.0001. COVID-19 group III-APS: *p* < 0.0001. COVID-19 group II-COVID-19 group III: *p* = 0.04.
Male	13	37	58	16	

* Kruskal–Wallis test with Dunn’s multiple comparison test was used for age; ** χ^2^ test was used for sex.

**Table 2 ijms-24-00211-t002:** The prevalence of positive aPL in sera samples from HCPs before and after vaccination (group I), COVID-19 patients hospitalized in the ICU (group II), COVID-19 patients hospitalized in the non-ICU (at three time points: admission, hospital discharge, 3-month follow-up after hospital discharge) (group III), and APS patients (group IV).

	Group I		Group II	Group III			Group IV
	HCPs (n = 58)		COVID-19 Patients Hospitalized in ICU (n = 45)	COVID-19 Patients Hospitalized in Non-ICU (n = 89)			APS Patients (n = 52)
Time Points	Before Vaccination	After Vaccination *	During Hospitalization	At Hospital Admission	At Hospital Discharge	3 Months after Discharge	
aPL	No. of positive (%)						
aCL IgG	0	1 (1.7)	7 (15.6)	10 (11.2)	26 (29.2)	13 (14.6)	24 (46.2)
aCL IgM	1 (1.7)	1 (1.7)	1 (2.2)	4 (4.5)	18 (20.2)	5 (5.6)	19 (36.5)
aCL IgA	0	0	0	3 (3.4)	0	0	1 (1.9)
anti-β2GPI IgG	5 (8.6)	7 (12.1)	3 (6.7)	10 (11.2)	7 (7.9)	5 (5.6)	22 (42.3)
anti-β2GPI IgM	1 (1.7)	1 (1.7)	0	1 (1.1)	1 (1.1)	3 (3.4)	9 (17.3)
anti-β2GPI IgA	1 (1.7)	1 (1.7)	5 (11.1)	4 (4.5)	4 (4.5)	4 (4.5)	7 (13.5)
aPS/PT IgG	0	0	1 (2.2)	0	0	0	22 (42.3)
aPS/PT IgM	0	0	0	5 (5.6)	5 (5.6)	3 (3.4)	19 (36.5)
aPS/PT IgA	0	0	5 (11.1)	1 (1.1)	3 (3.4)	1 (1.1)	15 (28.8)
at least one positive aPL	7 (12.1)	10 (17.2)	15 (33.3)	23 (25.8)	44 (49.4)	23 (25.8)	44 (84.6)
single aPL positivity **	6 (10.3)	9 (15.5)	12 (26.7)	18 (20.2)	35 (39.3)	17 (19.1)	20 (38.5)
double aPL positivity **	1 (1.7)	1 (1.7)	2 (4.4)	3 (3.4)	8 (9.0)	5 (5.6)	7 (13.5)
triple aPL positivity **	0	0	1 (2.2)	2 (2.2)	1 (1.1)	1 (1.1)	17 (32.7)

* Values comprise five consecutive tests after vaccination. ** Calculations comprise results of aCL, anti-β2GPI and aPS/PT of all isotypes.

**Table 3 ijms-24-00211-t003:** aPL dynamics in COVID-19 patients during infection with SARS-CoV-2, presented as a longitudinal trend.

Longitudinal Trends between Time Points at Admission, at Hospital Discharge, and at Follow-Up		aCL	Anti-β2GPI	aPS/PT
		No. of Samples (%)	No. of Samples (%)	No. of Samples (%)
Samples negative at all time points		44 (49.4)	70 (78.7)	82 (92.1)
Samples positive at least at one time point		45 (50.6)	19 (21.3)	7 (7.9)
Trend 1: Levels increased during hospitalization and persisted at follow-up	/ ‾	8 (17.8)	2 (10.5)	0
Trend 2: Levels increased during hospitalization and decreased at follow-up	/ \	25 (55.6)	1 (5.3)	1 (14.3)
Trend 3: Levels were negative during hospitalization and increased at follow-up	_ /	7 (15.6)	3 (15.8)	0
Trend 4: Levels decreased during hospitalization and remained negative at follow-up	\ _	3 (6.7)	8 (42.1)	3 (42.9)
Trend 5: Levels were positive at all three time points	‾ ‾	2 (4.4)	3 (15.8)	2 (28.6)
Trend 6: Mixed trend		0	2 (10.5)	1 (14.3)

**Table 4 ijms-24-00211-t004:** Longitudinal data from HCPs who developed aPL during the observation period after vaccination. Legend: ms—missing sample. Positive levels: aCL IgG ≥ 11, anti-β2GPI IgG ≥ 2.

HCPs	Autoantibodies that Tested Positive after Vaccination	Time Points					
		Before Vaccination	3 Weeks after Vaccination	3 Months after Vaccination	6 Months after Vaccination	9 Months after Vaccination	3 Weeks after the Third Dose
HCP-60	aCL IgG (AUG)	3	1	4	13	ms	ms
HCP-21	anti-β2GPI IgG (AUG)	1	2	1	2	2	2
HCP-65	anti-β2GPI IgG (AUG)	1	3	2	3	1	2

## Data Availability

All anonymized raw data are fully available upon reasonable request.

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
