# Peer review of "Longitudinal Analysis of Antiphospholipid Antibody Dynamics after Infection with SARS-CoV-2 or Vaccination with BNT162b2"

_ijms, 2022, doi:10.3390/ijms24010211_

Round 1

Reviewer 1 Report

Post the outbreak of 2019 novel Coronavirus, many researchers have published about vast number of biomarkers as being related to the infection with Covid-19 primarily based on small subset of observations and without solid background studies or in-depth studies. This is where the manuscript by Ogric et al stands out. Not only have they performed the current study in depth with appropriate controls, but also given credit and discussed the shortcomings of related studies previously published regarding transient expression of antiphospholipid antibodies (aPL) during SARS-Cov2 infection or after vaccination with the BioNTech Pfizer (BNT162b2) vaccine.

In the current study, the authors aim to investigate the expression of aPL in Covid-19 patients at various time-points of their infection and compare it with healthy individuals who received BNT162b2 and individuals with antiphospholipid syndrome (APS). Their in-depth longitudinal analysis reveals that although aPL is induced in few individuals post-vaccine and in patients with Covid-19, the persistence of aPL levels is never to the extent of APS patients.

Well written manuscript. The literature review is in-depth and well discussed. The Materials and Methods section is well described, and the approach is acceptable for the kind of study presented in this article.

Comments

1.     Was there any rationale behind selecting healthcare professionals as the vaccine received group and not any random healthy volunteers who received the same vaccine?

Author Response

  1. Was there any rationale behind selecting healthcare professionals as the vaccine received group and not any random healthy volunteers who received the same vaccine?

Thank you for your comment. We chose healthcare professionals because this was the group that was first enrolled in the vaccination program in our country and all samples before and after vaccination were easily collected at the Division of internal medicine.

Reviewer 2 Report

The authors reported the longitudinal study of aPL level within health care workers, COVID19 patients, and APS patients, showing that BNT162b2 mRNA vaccine may induce low level of aPL response, which not follow the pattern in APS patients. Minor comments as below:

1. line 105, remove the comma after ‘To’

2. line 122, should start with ‘2.1’, not ‘2.2’

3. Table 1/2/3/4, all could be converted to figures to increase the readability, and facilitate the data presentation

4. The age bias needs proper discussion in the last section

Author Response

  1. line 105, remove the comma after ‘To’

Thank you for this correction.

  1. line 122, should start with ‘2.1’, not ‘2.2’

Thank you for this correction.

  1. Table 1/2/3/4, all could be converted to figures to increase the readability, and facilitate the data presentation

Thank you for the good suggestion. We will pay special attention when proof reading for the style of the Table and ask the editor for corrections when needed.

  1. The age bias needs proper discussion in the last section

We agree with your comment and are aware that age bias should be discussed. We added the following paragraph to the manuscript:

''While APS usually affects young people, because the first vascular event usually occurs in young adults and rarely in people older than 60 years, the frequency of aPL positivity in the general population is known to increase with age [37]. Therefore, the different age distribution of our patient groups (COVID-19 patients were older, whereas APS patients were younger) not only reflects etiologies of these conditions and a possible bias in our study, but also confirms that the frequency of aPL positivity is much higher in APS patients compared with COVID-19, although the APS group was the youngest one in this study.

Reviewer 3 Report

In this study, the authors investigate the dynamics of aPL in COVID-19 patients and in individuals (healthcare professionals - HCPs) after receiving vaccine. They also compare aPL levels and positivity with those found in patients with APS.

This is an interesting subject. However, some several points require attention and major revisions must be made before the manuscript can be reconsidered.

-                  Concerning the prevalence of apl in Covid-19, several bibliographic references are missing as for example those of Bertin et al (https://www.ncbi.nlm.nih.gov/pmc/articles/PMC7323091/#), and also the metanalysis showing that aPLs were detected in nearly half of patients with COVID-19, and that a higher prevalence of aPL was found in severe disease (Taha M, Samavati L. RMD Open 2021;7:e001580. doi:10.1136/rmdopen-2021-001580). In addition the persistence of apl after COVID-19 was also investigate in a recent study (https://www.researchgate.net/publication/365196162_True_Antiphospholipid_Syndrome_in_COVID-19_Contribution_of_the_Follow-up_of_Antiphospholipid_Autoantibodies

                   I do not understand the justification for comparing patients with COVID-19 / or vaccinated with APS patients. Clinical criteria and aPL persistence are necessary to attest APS diagnosis. It's just a way of giving an order of magnitude in terms of prevalence. Can the authors re-specify?                      As the question raised by the authors is important, it should be necessary to increase the cohort, in particular the numbers concerning patients with COVID-19 and vaccinated patients (to be doubled). Moreover, clinical data should be add for all patients and subjects, in particular those concerning potential thrombotic or obstetrical events                      Concerning the presence of apl in healthy subjects, it is generally recommended to use a threshold value at the 99th percentile and then the prevalence corresponds to i.e. 1% of the general normal population. In the study, the presence of apl in healthy subjects at a fairly high prevalence is intriguing. This prevalence may be related to the assay used. Positive results (in aCL and antiB2GPI) should be confirmed by another assay. Once again clinical status should be added                      The results of each isotype is not systematically precise in tables                    The figures relating in particular to the longitudinal follow-up could be better informative.  

Author Response

Concerning the prevalence of aPL in Covid-19, several bibliographic references are missing as for example those of Bertin et al (https://www.ncbi.nlm.nih.gov/pmc/articles/PMC7323091/#), and also the metanalysis showing that aPLs were detected in nearly half of patients with COVID-19, and that a higher prevalence of aPL was found in severe disease (Taha M, Samavati L. RMD Open 2021;7:e001580. doi:10.1136/rmdopen-2021-001580).

Thank you for your comment. We have updated the text in the 3rd paragraph of the introduction and added the proposed references on aPL prevalence in patients with COVID-19.

“Contrary, a study comparing moderate and severe form of COVID-19 disease that IgG aCL levels were highly and independently associated with disease severity [14].”

A large meta-analysis, comprising 21 studies and 1159 patients, published in April 2021, showed that nearly half of patients with COVID-19 were positive for at least one of the aPL. Most frequently reported aPL was LA. aPL were significantly more frequently reported in critically ill patients, and aPL were not significantly associated with disease outcomes like venous thrombosis, invasive ventilation and mortality [16].”

In addition, the persistence of aPL after COVID-19 was also investigate in a recent study (https://www.researchgate.net/publication/365196162_True_Antiphospholipid_Syndrome_in_COVID-19_Contribution_of_the_Follow-up_of_Antiphospholipid_Autoantibodies

Thank you for this excellent suggestion. We have updated the text in the 3rd paragraph of the discussion and added the proposed study on longitudinal measures of aPL in patients with COVID-19.

''Recently another group of investigators examined the persistence of aPL positivity after COVID-19.  Of the 45 aPL positive patients who were followed up for 12 weeks thirteen patients (28.9%) had at least one persistent aPL with a single positivity in 69.2%, double positivity in 15.4%, and triple positivity in 15.4% [33].''

I do not understand the justification for comparing patients with COVID-19 / or vaccinated with APS patients. Clinical criteria and aPL persistence are necessary to attest APS diagnosis. It's just a way of giving an order of magnitude in terms of prevalence. Can the authors re-specify?   

Agree. APS is a disease or condition that is difficult to diagnose. There is no specific diagnostic marker, so classification criteria have been proposed to recognize and treat patients with the proposed clinical and serological markers. To meet the laboratory classification criteria, at least one of the criteria aPL must be positive twice. Therefore, none of the single aPL is positive in 100% of patients and each patient has a different combination of aPL positivity. A patient can be either single/double or triple positive, they can have IgG or IgM antibodies. But as our study shows, the levels and frequency of positivity are much higher in APS compared to controls, COVID-19 patients or vaccinated individuals. Therefore, our main conclusion was that although some aPL may occur during infection with SARS-CoV-2 and may also be persistently positive, they are far from reaching the frequency and magnitude of positivity typical of APS.

As the question raised by the authors is important, it should be necessary to increase the cohort, in particular the numbers concerning patients with COVID-19 and vaccinated patients (to be doubled).

Thank you very much for this suggestion. We agree that it would have been important to increase our cohorts (HCPs, n=58, COVID-19 ICU, n=45, COVID-19 non-ICU n=89). We also agree that this is a potential limitation which we have emphasized in the discussion.  On the other hand, we would like to point out that several other recent studies have published their research in even smaller groups of patients. Serrano et al. comprised several studies in their review and the median number of included participants in the studies was 73 (range: 3 - 474 included patients).

Moreover, clinical data should be add for all patients and subjects, in particular those concerning potential thrombotic or obstetrical events    

Thank you very much for this suggestion. We have added additional data considering how clinical information were monitored and obtained in the method section as well as we have added new subheading regarding clinical data in the manuscript.

In Methods:

“Group II comprised 45 COVID-19 patients with a severe form of the disease who were treated in the ICU UMCL in Slovenia from May to December 2020 before the start of vac-cination. Their serum samples were collected on days 8-12 of their hospitalization. Clini-cal manifestations significant for APS were monitored.”

“Group III comprised 89 COVID-19 patients hospitalized in the non-intensive care unitsICU at CHC Pančevo from January 2021 to May 2021. Their serum samples were col-lected at admission, at the hospital discharge (days 5-29) and at 3-month follow-up after hospital discharge. Clinical data, comprising history of clinical manifestations significant for APS, were obtained as well as monitored during the observation period.”

In results:

“2.5. Clinical manifestations related to APS in vaccinated HCPs and COVID-19 patients

None of the HCPs who had positive levels of aPL experienced any thrombotic ad-verse events during the 1-year observation period.

Two COVID-19 patients (group II) hospitalized in the ICU had arterial thrombosis, ten had venous thrombosis, and four had both venous and arterial thrombosis. Six of them had positive levels of aPL on one occasion. One patient who experienced arterial and venous thrombosis was triple positive for aCL, anti-β2GPI and aPS/PT and died.

Four COVID -19 patients (group III), hospitalized in the non-ICU, had a history of ar-terial thrombosis, three had a history of venous thrombosis, and two had a history of ob-stetric complications relevant to the APS classification criteria. None of these nine patients was classified as an APS patient, and none of them had positive aPL values twice during our observation. Moreover, none of them experienced a new thrombotic event during our observation. One patient experienced arterial thrombosis and microthrombosis during hospitalization in the non-ICU and was triple aPL positive at admission and remained double positive at hospital discharge and 3-month follow-up. This patient thus met both clinical and laboratory criteria for definite APS.”

Concerning the presence of aPL in healthy subjects, it is generally recommended to use a threshold value at the 99th percentile and then the prevalence corresponds to i.e. 1% of the general normal population. In the study, the presence of aPL in healthy subjects at a fairly high prevalence is intriguing. This prevalence may be related to the assay used. Positive results (in aCL and antiB2GPI) should be confirmed by another assay.

We agree with this comment. The cut-off for our in-house ELISA was set at 99th percentile as recommended by the international classification criteria for APS. This was performed in 222 healthy subjects mean age 43y, range 19-65) and the publication of our in-house ELISAs are cited in the methods as follows:

“aPL were determined using in-house ELISAs according to previously described pro-tocols, specifically aCL ELISA [48], anti-β2GPI ELISA, aPS/PT ELISA [49]. Values above the 99th percentile of the healthy control population were considered positive, specifically for aCL ≥11AU, for anti-β2GPI ≥2 AU and for aPS/PT ≥5AU.”

Regarding the reviewers' concern that the prevalence of aPL in our health professionals is higher than the expected 1% in the general population, we have some explanations. First, health care professionals included in this study were older compared with the healthy blood donors tested in previous studies to define the cut-off at the 99th percentile. The frequency of aPL positivity in the general population is known to increase with age. Second, health care professionals are actually not blood donors. Although they are generally healthy and reported having no rheumatic autoimmune diseases, some of them had other conditions (7 with arterial hypertension, 4 with asthma, 2 with Hashimoto`s disease, 1 with liver autoimmune disease...).  We specifically checked whether these diseases correlated with aPL but could not detect a pattern.

Once again clinical status should be added           

Thank you. Please see our previous answers to this issue

The results of each isotype are not systematically precise in tables 

We appreciate your insightful suggestion and agree that it would be useful to demonstrate. If we understand you correctly, this issue was raised with respect to Table 3, since the positivity of each isotype is already indicated in Table 2 and Table 4 for each group. In our opinion, adding these data to Table 3 would extremely expand the content and we are concerned that adding these data would result in less comprehensibility. Also, data for each individual isotype investigated separately is shown in Figure 1 for easier presentation. As explained later we split the Figure 1 into three individual Figures and we are now confident that the information is now much clearer.

The figures relating in particular to the longitudinal follow-up could be better informative.

Agreed. The Figure 1 showing longitudinal follow-up is very complex and difficult to understand and we apologize if our original figure was not clear and informative. We have split the Figure 1 into three individual Figures and added a several sentences to address this.

Figure 1. Longitudinal trends of aCL, anti-β2GPI, and aPS/PT, IgG, IgM, and IgA during infection with SARS-CoV-2 in COVID-19 patients hospitalized in the non-ICU (group III) in three time points (1 - at admission, 2 - at hospital discharge, 3 - at 3- month follow-up after hospital discharge). Values above the green line (99th percentile of healthy blood donors) are considered positive.

Figure 2. Longitudinal trends of aCL, anti-β2GPI, and aPS/PT, IgG, IgM, and IgA before (0) and after vaccination (time points: 1 - 3 weeks after vaccination, 2 - 3 months after vaccination, 3 - 6 months after vaccination, 4 - 9 months after vaccination, and 5 - after the third – booster dose). Values above the green line (99th percentile of healthy blood donors) are considered positive.

Figure 3. Levels of aCL, anti-β2GPI, and aPS/PT, IgG, IgM, and IgA in HCPs after vaccination (group I), COVID-19 patients hospitalized in the ICU unit (group II), COVID-19 patients hospitalized in non-ICU (group III) (at hospital discharge), and APS patients (group IV). Legend: Kruskal Wallis test with Dunn`s multiple comparison test (significances in graphs) was used for the data shown in C. Values above the green line (99th percentile of healthy blood donors) are considered positive

Round 2

Reviewer 3 Report

thank you for your answers and comments